# *In Vitro* Osteogenesis Study of Shell Nacre Cement with Older and Young Donor Bone Marrow Mesenchymal Stem/Stromal Cells

**DOI:** 10.3390/bioengineering11020143

**Published:** 2024-01-31

**Authors:** Bridget Jeyatha Wilson, Heather Elizabeth Owston, Neelam Iqbal, Peter V. Giannoudis, Dennis McGonagle, Hemant Pandit, Lizymol Philipose Pampadykandathil, Elena Jones, Payal Ganguly

**Affiliations:** 1Division of Dental Products, Department of Biomaterial Science and Technology, Biomedical Technology Wing, Sree Chitra Tirunal Institute for Medical Sciences and Technology, Thiruvananthapuram 695 012, India; 2Leeds Institute of Rheumatic and Musculoskeletal Medicine, University of Leeds, Leeds LS9 7JT, UKd.g.mcgonagle@leeds.ac.uk (D.M.); h.pandit@leeds.ac.uk (H.P.);; 3School of Chemical and Process Engineering, University of Leeds, Leeds LS2 9JT, UK; 4Leeds Orthopaedic & Trauma Sciences, Leeds General Infirmary, University of Leeds, Leeds LS2 9JT, UK; 5Leeds Musculoskeletal Biomedical Research Centre, Chapel Allerton Hospital, Leeds LS7 4SA, UK

**Keywords:** bone void filling cement, shell nacre cement, bone marrow mesenchymal stem cells: age-related changes:proliferation, osteogenesis

## Abstract

Bone void-filling cements are one of the preferred materials for managing irregular bone voids, particularly in the geriatric population who undergo many orthopedic surgeries. However, bone marrow mesenchymal stem/stromal cells (BM-MSCs) of older-age donors often exhibit reduced osteogenic capacity. Hence, it is crucial to evaluate candidate bone substitute materials with BM-MSCs from the geriatric population to determine the true osteogenic potential, thus simulating the clinical situation. With this concept, we investigated the osteogenic potential of shell nacre cement (SNC), a bone void-filling cement based on shell nacre powder and ladder-structured siloxane methacrylate, using older donor BM-MSCs (age > 55 years) and young donor BM-MSCs (age < 30 years). Direct and indirect cytotoxicity studies conducted with human BM-MSCs confirmed the non-cytotoxic nature of SNC. The standard colony-forming unit-fibroblast (CFU-F) assay and population doubling (PD) time assays revealed a significant reduction in the proliferation potential (*p* < 0.0001, *p* < 0.05) in older donor BM-MSCs compared to young donor BM-MSCs. Correspondingly, older donor BM-MSCs contained higher proportions of senescent, *β*-galactosidase (SA-*β* gal)-positive cells (nearly 2-fold, *p* < 0.001). In contrast, the proliferation capacity of older donor BM-MSCs, measured as the area density of CellTracker^TM^ green positive cells, was similar to that of young donor BM-MSCs following a 7-day culture on SNC. Furthermore, after 14 days of osteoinduction on SNC, scanning electron microscopy with energy-dispersive spectroscopy (SEM-EDS) showed that the amount of calcium and phosphorus deposited by young and older donor BM-MSCs on SNC was comparable. A similar trend was observed in the expression of the osteogenesis-related genes BMP2, RUNX2, ALP, COL1A1, OMD and SPARC. Overall, the results of this study indicated that SNC would be a promising candidate for managing bone voids in all age groups.

## 1. Introduction

Bone is the only biomineralised composite with cellular compositions that can regenerate without scar formation. The orchestrated cellular actions of bone-forming osteoblasts, bone-resorbing osteoclasts, mechanosensory osteocytes, and bone-repairing and-regenerating bone marrow mesenchymal stem/stromal cells (BM-MSCs) control the mass, quality and strength of bone [1,2]. BM-MSCs are key players, and can self-renew and maintain bone homeostasis. BM-MSCs are multipotent and can differentiate into either osteoblasts, chondrocytes or adipocytes according to stimuli and demand [3,4,5]. Within the bone marrow (BM) compartment, BM-MSCs are actively involved in osteogenesis and ensure balanced and dynamic bone modelling and remodelling in healthy bones [6,7,8]. Unfortunately, age-related changes occur in the cellular and extracellular matrix of bone, influencing bone quality and increasing the susceptibility to fractures and bone loss [9,10]. Senescence, or the irreversible arrest of cellular growth and proliferation in bone, is elicited by oxidative stress, altered mitochondrial metabolism, aberrant epigenetic modification, DNA damage, telomere dysfunction, heterochromatin changes, mutation, and oncogene expression. All of these factors contribute to the aging of the bone microenvironment [11,12,13]. In particular, the BM-MSCs of osteoporosis and osteopenia patients exhibit reduced osteogenic potential and increased BM adiposity [14,15,16,17,18,19]. Furthermore, aging causes a decline in the ability of BM-MSCs to proliferate, form colonies and self-renew [11,20].

Clinically, osteoporosis or age-related fractures affect 1 in 3 women and 1 in 5 men over the age of 50 years [21]. A global study in 2019 reported a progressive increase in new fractures, with older age groups experiencing higher fracture incidence rates [22]. However, the availability of bone autograft is often a limiting factor, especially with older patients. The treatment strategies for managing bone loss and bone voids include augmentation techniques involving bone grafts or a range of bone cements or ceramics [23,24,25,26,27]. Polymethyl methacrylate (PMMA) is widely used in various clinical settings, such as in the treatment of fragility fractures [28], vertebral compression fractures [29], revision arthroplasty defects [30,31,32], and bone voids after infection [33] or tumour resections [34]. While PMMA offers the required moldability and mechanical strength, it lacks essential biological properties and resorbability [35,36,37]. On the other hand, alternative biomaterials like calcium phosphate and calcium sulphate exhibit improved biological properties but often lack the necessary mechanical properties [23,25,38].

With these issues in mind, shell nacre cement (SNC) was developed as a novel bone void-filling cement with improved biological and physicochemical properties. The major composition of the cement is shell nacre, the inner lustrous layer of pearl oyster shell, which has demonstrated osteogenic, anti-osteoporotic and angiogenic potentials [39,40,41]. Another prime constituent is ladder-structured siloxane methacrylate (SNLSM2), an inorganic–organic hybrid resin known for its low shrinkage and good mechanical properties [42,43]. In combination with SNLSM2, SNC exhibited higher compressive strength, low polymerization shrinkage, minimal exothermic behaviour and no cytotoxicity against the L929 cell line. [44]. However, the osteogenic potential of SNC has not been investigated. As it has been reported that, during aging, BM-MSCs exhibit a shift towards adipogenesis rather than osteogenesis [14,15,16,17,20,45], and given that more than 50% of orthopaedic procedures are performed on older patients [46], evaluating the osteogenic potential of SNC with BM-MSCs from older donors is essential. Previous *in vitro* osteogenesis studies of other biomaterials such as PCL nanofibrous mat, graphene-incorporated methacrylate gelatin, and rattan wood have been conducted with BM-MSCs from older donors [47,48,49]. However, none of these studies have investigated the senescence status of the older donor BM-MSCs prior to scaffold seeding. This *in vitro* study is therefore the first, in which a comparison of the proliferation capacity and senescence of human BM-MSCs from older and young donors and an evaluation of the osteogenic potential when cultured on a bone void-filling cement SNC were performed.

The main objective of this study was to investigate the *in vitro* osteogenic potential of SNC with iliac crest BM-MSCs from older and young donors. This study included evaluations of the colony-forming capacity, proliferative potential and senescence of older and young donor BM-MSCs prior to seeding on scaffold. Following the direct and indirect cytotoxicity studies of the SNC samples with BM-MSCs, CellTracker^TM^ Green staining was performed and the proliferation of BM-MSCs on SNC was investigated using confocal microscopy. Additionally, the influence of SNC following osteoinduction was assessed using scanning electron microscopy with energy-dispersive spectroscopy (SEM-EDS) and qPCR. The flowchart below (Figure 1) illustrates the experimental workflow used for this study.

## 2. Materials and Methods

### 2.1. Shell Nacre Cement (SNC) Samples

The development and physico-chemical characterization of SNC have been outlined previously [44]. In brief, shell nacre powder was processed from the shells of *Pinctada fucata*. Subsequently, shell nacre-integrated ladder-structured siloxane methacrylate resin (SNLSM2) was synthesised. SNC was formulated using shell nacre powder (72%), SNLSM2 resin (12%) and other ingredients such as triethylene glycol dimethacrylate (12%), fumed silica (3%), benzoyl peroxide—0.7% (initiator), dimethylaminophenyl ethanol—0.4% (activator), traces of butylated hydroxy toluene and 4-methoxy phenol [44]. Cured cylindrical SNC samples with the dimensions of 10 mm diameter × 1 mm height and 4 mm diameter × 1 mm height were prepared and sterilised using ethylene oxide (EO) at Sree Chitra Tirunal Institute for Medical Sciences and Technology, Thiruvananthapuram, India. Prior to culturing with BM-MSCs, SNC cement samples were pre-conditioned in StemMACS (SM) medium with 1% Pencillin-Streptomycin (P/S) (Miltenyi Biotec, Bisley, UK) for 72 h at 37 °C and 5% CO_2_.

### 2.2. Ethical Approval and Donor Details

Ethical approval was obtained from NREC Yorkshire and Humberside National Research Ethics Committee (06/Q1206/127 and 18/YH/0166) to collect human donors’ bone marrow aspirate (BMA) samples.

BMA was collected in ethylenediaminetetraacetic acid (EDTA) tubes to prevent coagulation and harvested from *n* = 10 donors between 21 and 64 years old. Out of these 10 donors, *n* = 3 were categorised as ‘young’ donors aged below 30 (ranging between 21–26 years old, 3 males), *n* = 3 were classified as ‘older’ donors aged above 55 years old (between 58–64 years old, 2 males, 1 female) and *n* = 4 were classified as ‘middle-aged donors’, with their age ranging from 30 to 50 years old (2 males, 2 females). The collected BMA was processed as per established methods [50,51]. Briefly, BMA was passed through a 70 µm cell filter (ThermoFisher Scientific, Loughborough, UK). It was then treated with ammonium chloride lysis buffer for red blood cell removal and washed with PBS. Finally, the mononuclear cell fraction was frozen at −80 °C.

Frozen vials were defrosted in complete DMEM mediumcontaining 10% FBS and 1% P/S (ThermoFisher Scientific, Loughborough, UK), then cultured in SM medium, expanded and tracked for growth kinetics. On reaching confluency, the cells were trypsinised and used for further experiments. 

### 2.3. Direct Cytotoxicity Studies

The direct contact cytotoxicity of the cured SNC samples (diameter 4 mm × 1 mm) was investigated using pooled BM-MSCs from middle-aged donors. The SNC samples were secured to the surface of a 6-well plate (Corning, NY, USA) using SteriStrip^TM^ tape to prevent the disruption of the BM-MSC monolayer. The monolayers of BM-MSCs and SteriStrip^TM^ (10 mm length) (MR1547, 3M^TM^, Medisave, Dorset, UK) served as the controls. After 1 and 4 days, the cellular response around the material was observed for morphological changes, the zone of lysis, vacuolization, membrane disruption, and cellular detachment.

### 2.4. Indirect Cytotoxicity Studies—XTT Assay

The pooled BM-MSCs from middle-aged donors were used for indirect cytotoxicity studies. The SM medium was used as the base extraction medium, and the extracts were prepared after 3 days, 7 days, and 14 days of contact with the sterile SNC samples based on ISO 10993-12:2021 [52]. Briefly, 6 mL of SM medium was dispensed per well of a 6-well plate, where the SNC samples (diameter 4 mm × 1 mm) were placed. Subsequently, medium extractions were collected (620 μL in triplicate) at the aforementioned time points and stored at −80 °C [53]. Cell viability and proliferation measurements were performed via XTT assay, as per the manufacturer’s protocols.

#### 2.4.1. Cell Viability (24 h Exposure)

BM-MSCs (10^4^ cells) were seeded per well in a 96-well plate in SM medium. After 24 h of incubation, the SM medium was replaced with 200 µL of the SNC sample extract, 10% DMSO SM medium control, and SM medium control. After 24 h of exposure, the treatment medium was replaced with 100 μL of complete DMEM and 50 µL of the XTT (sodium 3′-[phenylaminocarbonyl)-3,4-tetrazolium]-bis (4-methoxy-6-nitro) benzene sulfonic acid hydrate) solution (cell proliferation kit II (XTT), Roche, Munich, Germany) and incubated in the dark for 4 h at 37 °C. The yellow-coloured XTT was reduced to orange-coloured formazan in the presence of metabolically active cells. The orange colour developed by the metabolically active cells was read at 450 nm and 630 nm (reference wavelength) using a spectrophotometer, and the value of the reference wave at 630 nm was deducted from 450 nm to give the final optical density (OD). The percentage metabolic activity of the cells was calculated using the following formula: (OD test/OD SM Medium control) ×100.

#### 2.4.2. Cell Proliferation (4 Days Exposure)

BM-MSCs (500 cells) were seeded per well in a 96-well plate, for 24 h in SM medium. After 24 h, the SM medium was replaced with 200 µL of the treatment medium containing either the SNC sample extract, SM medium with 10% DMSO, or the control SM medium. Cells were then cultured for 4 days in the treatment medium. After 4 days, the XTT assay was carried out as described above (Section 2.4.1) and analysed to quantify the cell proliferation.

### 2.5. Characterization of Older and Young Donor BM-MSCs

Prior to seeding on SNC samples, the following population doubling (PD) time assay, colony-forming unit fibroblast (CFU-F) assay and senescence-associated beta-galactosidase (SA-*β*-gal) assay were performed.

#### 2.5.1. Calculation of Population Doubling Time Assay

The PD time of the BM-MSCs from individual donors was studied from passage 0 (p0) to passage 3 (p3) using the formula PD = log_2_(cells trypsinised/cells seeded), and the average cumulative PD of young (*n* = 3) and older donor BM-MSCs (*n* = 3) was compared between the BM-MSCs.

#### 2.5.2. Colony-Forming Unit-Fibroblast (CFU-F) Assay

As per the standard CFU-F assay protocol [11,51,54], BM-MSCs (2 × 10^3^) of the same passage (p2 to p3) were seeded in duplicate into 100 mm Petri dishes (Corning, NY, USA) containing 10 mL of SM medium and cultured for 2 weeks. The medium was changed entirely after 48 h, and half medium changes were subsequently performed twice a week. After 2 weeks of culture, the medium was removed, the dishes washed in phosphate-buffered saline (PBS, Life Technologies, Paisley, UK), and the cells fixed with 3.7% formaldehyde (Fisher Scientific, Loughborough, UK) and stained with 1% (*w*/*v)* methylene blue (Sigma, Dorset, UK) for 30 min. Subsequently, after the final distilled water wash, the plates were allowed to dry and then scanned at 1200 dpi for the further analysis of colonies. The colony area and colony-integrated density (ID) were analysed using Image J, as previously described [11]. In brief, the scanned images were converted into 8-bit images and calibrated. Each colony was selected manually, and colony area and density were calculated simultaneously by the software.

### 2.6. Studies with Pooled Older and Young Donor BM-MSCs

#### 2.6.1. Cell Morphology

After studying the colony-forming ability of individual donor BM-MSCs, all the older donor BM-MSCs (*n* = 3) were pooled together (ratio 1:1:1) at passage 3, then cultured and expanded. Similarly, the young donor BM-MSCs (*n* = 3) were pooled, cultured and expanded. Both the pooled older and young donor BM-MSCs were observed using phase-contrast microscopy (EVOS^®^, FL, Thermo Fisher Scientific, Waltham, MA, USA) to assess their morphology.

#### 2.6.2. Senescence-Associated Beta-Galactosidase (SA-*β*-Gal) Assay

Pooled older and young donor BM-MSCs were plated separately in duplicate at 4 × 10^4^ per well in a 6-well plate overnight for adhesion in 1.5 mL of SM medium. After 24 h, the medium was removed, and the cells were washed in PBS. The cells were fixed using fixation buffer (1.5 mL) and incubated for 6 to 7 min at room temperature. Following a PBS wash, the cells were stained with X-Gal staining mixture (1 mL), prepared as per the manufacturer’s protocol (Senescence Cells histochemical staining kit, Sigma Aldrich, St. Louis, MO, USA). In brief, 10 mL of the staining mixture included 400 mM of Potassium Ferricyanide (125 µL), 400 mM of Potassium Ferrocyanide (125 µL), X-gal solution (40 mg/mL, 1 mL) and ultrapure water (8.5 mL). The dishes were sealed in parafilm and incubated at 37 °C without CO_2_ overnight. The next day, the cells were observed for the presence of blue (or SA-*β*-gal positive cells) under light microscopy, and the cells were counted using Image J software v1.52 [53].

#### 2.6.3. Confocal Microscopy of BM-MSC Attachment and Proliferation on SNC 

Both pooled older donor and young donor BM-MSCs (3 × 10^4^ cells) were loaded separately onto preconditioned SNC samples (10 mm diameter × 1 mm height) in triplicate in 24-well plates and incubated for 1 h. After the incubation, the SNC samples were moved to a low-attachment 24-well plate (Corning, NY, USA) and cultured for 7 days in SM medium. On day 7, the viability of the BM-MSCs on the SNC samples was assessed. CellTracker^TM^ Green (Thermo Fisher Scientific, Waltham, MA, USA) was prepared in serum-free DMEM medium, to a final working concentration of 0.7 µM and the samples were stained for 30 min at 37 °C and 5% CO_2_. Subsequently, the samples were imaged using confocal microscopy (Leica TCS SP8 confocal microscope, Leica Microsystems, Wetzlar, Germany). The cells present in each image (*n* = 3) of the SNC loaded with young and older donor BM-MSCs were counted using Image J software v1.52 and the average number of cells in each scaffold were calculated and normalised to the surface area. 

#### 2.6.4. *In Vitro* Osteogenic Potential of BM-MSCs Cultured on SNC

Both the pooled young and older donor BM-MSCs were separately seeded on the preconditioned triplicate SNC samples (10 mm diameter × 1 mm height) (3 × 10^4^ cells per sample) in 24-well plates and incubated for 1 h. After the incubation, the SNC samples were moved to a low-attachment 24-well plate and cultured for 7 days with SM medium, after which the SM medium was replaced with osteo-differentiation medium (OsteoDIFF Medium, Miltenyi Biotec, Bisley, UK) for a further 14 days; a half medium change was performed twice weekly [53]. At the end of the culture (day 14 of osteoinduction), the mineral deposition was studied by SEM-EDS, as previously described [55]. The control group comprised triplicate SNC samples without cells that were similarly treated with the SM medium for 7 days and the osteo-differentiation medium for 14 days. At the end of the culture, the samples were washed thrice with PBS and fixed with 3.7% paraformaldehyde. Following dehydration in increasing ethanol concentrations (30% to 100%), the samples were then evaluated using SEM-EDS. In brief, EDS mapping was performed to a depth of 2 µm to enable the measurement of the calcium/phosphorus signals in the neighbourhood of cells and near the SNC surface. The spectrum with the mass (%) values of calcium and phosphorus was obtained for each replicate sample and the values were normalised to the average mass (%) of calcium and phosphorus present in the control SNC samples and compared between young and older donor BM-MSCs.

#### 2.6.5. Gene Expression

Gene expression was performed to compare young and older donor BM-MSC osteogenic progression after days 7 and 14 of osteoinduction on SNC. Pooled young and older donor MSCs (1 × 10^6^ cells per sample) were separately seeded on duplicate SNC scaffolds, as described in 2.6.4. At the end of the culture, the SNC samples were processed for gene expression analysis, as previously described [47]. In brief, the samples were treated with 350 µL of Buffer RL to lyse the cells; strong agitation using vortex over ice was used to ensure the removal of all the cells with their genetic contents and to prevent RNA degradation. RNA was extracted according to the manufacturer’s protocol (Total RNA purification kit, Norgen Biotek, Canada), reverse transcription was performed using RT master mix (Fluidigm) and the cDNA was pre-amplified with the pooled Taqman assays, as previously described [47,53,56]. Gene expression was performed for genes related to MSC osteogenesis (day 14 samples) and cell cycle regulation (day 7 samples), including bone morphogenetic protein 2 (BMP2), RUNT-related transcription factor X2 (RUNX2), alkaline phosphatase (ALP), collagen type I Alpha 1 chain (COL1A1), Osteomodulin (OMD), secreted protein acidic and rich in cysteine (SPARC), and p16, p21 and p53, respectively. The complete list of Taqman probes can be found in the Appendix A. Hypoxanthine phosphoribosyl transferase1 (HPRT1) was used as a housekeeping gene [53]. 

### 2.7. Statistical Analysis

All statistical analysis were performed using GraphPad Prism software (version 9.5.1). Shapiro–Wilk and Kolmogorov–Smirnov tests were used to assess the normality of the data. If the data were not normally distributed, a Mann–Whitney Test was used to compare between the young and older donor groups; data were represented as median (interquartile range (IQR)). For normally distributed data, the unpaired *t*-test or ordinary ANOVA (analysis of variance) was used to compare the mean of the data, and the data were presented as mean ± standard deviation. The differences between the mean were statistically significant only if *p* ≤ 0.05. The symbols used to indicate the level of significance were as follows: ns—*p* > 0.05, *—*p* ≤ 0.05, **—*p* < 0.01, ***—*p* < 0.001, ****—*p* < 0.0001. 

## 3. Results

### 3.1. Non-Cytotoxic Nature of SNC with BM-MSCs

Direct contact cytotoxicity studies of the SNC sample with BM-MSCs (Figure 2a–f) revealed no signs of cytotoxicity for up to 4 days. The BM-MSC-only controls exhibited a classical spindle-shaped morphology (Figure 2a,b). After contact with the SNC samples for 1 and 4 days (Figure 2c,d), the cells retained their spindle-shaped morphology, similar to the BM-MSC-only control. BM-MSCs were found in very close contact and attached to the SNC samples. Likewise, the SteriStrip^TM^ controls (Figure 2e,f) showed no features of cytotoxicity, and the cell density was similar to the SNC samples. These observations indicated the non-cytotoxic nature of the SNC material with human BM-MSCs.

Indirect cytotoxicity studies were carried out using sample extracts from the SNC after incubation in SM medium for 3 days (3D), 7 days (7D), and 14 days (14D). The cell viability of the BM-MSCs was measured after 24 h of exposure, a period of time that is too short for cells to divide, and the cell proliferation was measured after 4 days of contact with the aforementioned extracts (the period sufficient for two MSC divisions) [57]. In the XTT assay, the yellow-coloured XTT is reduced to orange-coloured formazan by metabolically active cells and the orange colour developed is read at 450 nm using a spectrophotometer; the metabolic activity therefore corresponds to the cell viability (24 h) or cell proliferation (4 days). The cell viability measurements (%) (Figure 2g) of the BM-MSCs after contact with the extracts SNC 3D and SNC 14D were 115.1 ± 0.2% and 111.7 ± 3.2%, respectively, which were significantly (*p* < 0.001, *p* ≤ 0.05) higher than the SM Medium control (103.5 ± 3.2%); this indicates that there is some pro-survival activity present in the extracts. The cell viability after contact with the sample SNC 7D was 103.4 ± 2.6%, which was not significantly different to the SM medium control. As expected, the 10% DMSO medium resulted in 42.8 ± 3.8% cell viability, significantly lower than the SM Medium control (*p* < 0.0001).

The cell proliferation measurements (%) (Figure 2h) of the BM-MSCs after contact with SNC 3D, SNC 7D and SNC 14D were 88.6 ± 3.7%, 107.2 ± 8.9% and 93.6 ± 3.1%, respectively, which were similar to that of the SM medium control (100.0 ± 11.3%). As anticipated, the 10% DMSO medium displayed 43.7 ± 0.4% cell proliferation, which was significantly lower than the SM medium control (*p* < 0.0001). Therefore, indirect assays confirmed and extended our findings obtained using direct assays.

### 3.2. Lower Proliferation Capacity and Higher PD Time of Older Donor BM-MSCs

Upon visual inspection of the stained CFU-F plates (Figure 3a), the colonies of the young donor BM-MSCs appeared larger and denser when compared to the colonies of the BM-MSCs from older donors. Quantification was carried out, and the median (IQR) colony area of 34.3 (22.3) mm^2^ (Figure 3b) and median colony-integrated density (ID) of 5,050 (3,301) ID (Figure 3c) for the young donor BM-MSCs were significantly higher (*p* < 0.0001 for both colony area and density) when compared to the median colony area of 22.2 (14.8) mm^2^ (Figure 3b) and the median colony ID of 3,810 (2,539) ID (Figure 3c) for the older donor BM-MSCs.

Both the young and older donor BM-MSCs were cultured from p0 to p4. The PD time was calculated from the estimated number of cells seeded and trypsinised. The cumulative PD concerning the cumulative days was calculated for all the BM-MSCs. The average cumulative PD time of the BM-MSCs from older donors (Figure 3d) was 3.7 ± 0.6 days, which was significantly higher (Unpaired *t*-test *p* < 0.05) than that of the BM-MSCs from young donors (2.0 ± 0.6 days).

### 3.3. BM-MSCs of Older Donors Exhibited Senescent Features

Microscopic images of the pooled young donor BM-MSCs (Figure 4a) demonstrated the normal elongated spindle-shaped morphology, whereas the pooled older donor BM-MSCs exhibited a large, flattened shape (Figure 4b). The pooled young and older donor BM-MSCs were subjected to the SA-*β* Gal staining procedure, and the stained images are shown in Figure 4c,d. The percentage of blue-coloured *β*-galactosidase-positive cells in the pooled young donor BM-MSCs (35.3 ± 3.8%) was significantly lower (*p* < 0.001) than in the pooled older donor BM-MSCs (65.0 ± 3.6%) (Figure 4e). Altogether, the results from these experiments confirm that the older donor BM-MSCs were significantly deficient in proliferation and had a higher number of senescent cells than the young donor BM-MSCs.

### 3.4. Viability and Proliferation of Young and Older Donor BM-MSCs on SNC

Both sets of SNC samples seeded with young donor and older donor BM-MSCs were stained with CellTracker^TM^ Green on day 7 of culture. Confocal imaging confirmed the presence of viable cells on SNC. Both the young (Figure 5a) and older donor BM-MSCs (Figure 5b) demonstrated BM-MSC attachment, which was seen as a distinct layer of cells over the SNC samples. The average number of young donor BM-MSCs present in the SNC was 146 ± 37 cells/cm^2^, whereas the count of older donor BM-MSCs was 178 ± 49 cells/cm^2^, showing no statistically significant difference. (Figure 5c). These data indicated that in contrast to culturing on plastic, culturing on SNC potentially improved the proliferation capacity of older donor BM-MSCs, which performed similarly to young donor BM-MSCs. 

### 3.5. In Vitro Osteogenic Potential of SNC 

The *in vitro* mineralisation of BM-MSCs grown on the surface of SNC for 14 days in the osteogenic medium was investigated using SEM-EDS (Figure 6). SNC samples without the cells served as the material control and underwent the same treatment medium as the SNC-cultured BM-MSCs. Mineralisation was observed on SNC samples seeded with young donor BM-MSCs, as evidenced by the presence of overlapping calcium and phosphorus deposition compared to the SNC-only control. Similarly, older donor BM-MSCs on SNC exhibited mineralization, with the proof of overlapping calcium and phosphorus deposits. Figure 6B shows the normalised mass (%) of calcium and phosphorus on the BM-MSC seeded scaffolds. The normalised mass (%) of calcium deposited by older donor BM-MSCs (10.4 ± 5.6%) was slightly higher than young donor BM-MSCs (9.0 ± 0.3%), but the difference was not statistically significant. Similarly, the normalised mass (%) of phosphorous deposited by the older donor BM-MSCs (9.1 ± 1.2%) was higher than in the young donor BM-MSCs (6.8 ± 0.3%), but again the difference was not statistically significant. Overall, both the young and older donor BM-MSCs exhibited mineralisation on SNC, as observed with the presence of calcium and phosphorus deposition, providing evidence for the osteogenic potential of SNC. 

### 3.6. Gene Expression Studies

A gene expression study was conducted in order to validate our mineralisation findings. On day 14 of osteoinduction, the relative gene expression of the osteogenesis-related genes BMP2, RUNX2, ALP, COL1A1, OMD and SPARC in older donor BM-MSCs cultured on SNC was similar to that of young donor BM-MSCs (Figure 7A). On day 7 of osteoinduction, we measured the relative gene expression of the cell cycle-related genes p16, p21 and p53, and this was comparable between older and young donor BM-MSCs cultured on SNC (Figure 7B).

## 4. Discussion

In this study, we demonstrated the non-cytotoxic nature of SNC samples using BM-MSCs from middle-aged donors. The PD time and CFU-F assays revealed a reduced proliferative capacity in BM-MSCs from older donors compared to those from young donors. The pooled BM-MSCs from older donors showed a typical large, flattened morphology, while the BM-MSCs from young donors exhibited a common spindle-shaped morphology. Correspondingly, the SA-*β* gal assay revealed a higher proportion of *β*-gal-positive cells in older donor BM-MSCs compared to BM-MSCs from young donors, indicating significantly higher senescence levels in older donor BM-MSCs. However, when grown on SNC, the pooled BM-MSC cultures from young and older donors performed similarly in terms of viability and proliferation. Furthermore, following osteoinduction on SNC, BM-MSCs from older donors exhibited mineralization, with evidence of calcium and phosphorus deposition, along with the expression of osteogenesis-related genes; this is similar to what was observed in BM-MSCs from young donors. Once again, in the presence of SNC, BM-MSCs from older donors performed comparably to BM-MSCs from young donors, regardless of their initial senescence status.

Cytotoxicity experiments were conducted using BM-MSCs from middle-aged donors. These were performed to preserve precious BM-MSCs from older donors to be used in all the subsequent experiments. The comparisons in the direct cytotoxicity assays were performed between SNC samples secured with SteriStrip^TM^, SteriStrip^TM^ alone and cells alone. There were no visual differences between SteriStrip^TM^ alone and SteriStrip^TM^ + SNC, confirming the non-toxic nature of the SNC. In indirect assays, SteriStrip^TM^ was not required to secure the scaffolds in place; instead, SNC extracts were used, and the controls included the extraction medium alone (SM medium) and the medium containing DMSO (toxic to cells). The indirect assays showed no inhibition of BM-MSC proliferation with the addition of the 3-day SNC extracts, as well as longer extracts with a longer time (7 days and 14 days). These results were in accord with a previous study showing the non-cytotoxicity of SNC using the L929 cell line [44]. Notably, the cell viability (%) increased after contact with SNC, which may have been due to the elution of the shell nacre powder from the SNC cement samples. This finding is consistent with the results reported by Ruan et al., where the addition of nacre improved the proliferation of MC3T3-E1 cells [58]. 

In the present study, BM-MSCs collected from donors under 30 years of age were categorised as young, while those obtained from donors over 55 were classified as older. This classification considered the ongoing development of bone until the mid-twenties [59] and the declining proliferative capacity of BM-MSCs from the fifth decade [60]. Many studies with large patient cohorts have consistently reported that the proliferation and osteogenesis of older donor BM-MSCs decrease with increasing age [11,61,62,63,64]. As well, these and other studies have reported that donor variation within the same age group does exist and that it can be attributed to the donor’s ‘biological age’, rather than their ‘chronological age’. For this reason, in the present study, we did not use single cases of young and older donor MSCs. Based on previous experience with BM-MSC proliferation and senescence assays [61,65], the young and older donor groups consisted of *n* = 3 donors each, and they were at least 3 decades apart (young less than 30 years and older above 55 years). Indeed, by using these group sizes, we were able to confirm significant differences in the cell proliferation and senescence. The cumulative PD time of the older BM-MSCs was longer than that of the young BM-MSCs, consistent with previous studies reporting a longer PD time for older donor BM-MSCs [20,66]. The colony analysis of the CFU-F assay revealed a significant reduction in the colony area and integrated density, indicating the decreased proliferation potential of older donor BM-MSCs. These findings align with earlier reports of the reduced proliferative capacity of BM-MSCs from older donors [11,45,62,63,65]. 

To minimise donor variation within the groups, we separately ‘pooled’ (combined) BM-MSCs from the same young donors (*n* = 3) and older donors (*n* = 3). Alterations in cell morphology is a sign of senescence [67], and senescence was further assessed using the SA-*β* gal assay, a widely accepted procedure for cellular senescence [68]. *β*-galactosidase is a marker of cellular senescence, and *β*-gal-positive cells exhibited the senescence-like shortening of telomeres and the higher expression of the phosphorylated inhibitor of cyclin-dependent kinase 4A (p16^INK4a^) [69]. P16, a senescent marker [70] expressed in aging bone [71], collaborates with p21 (cell cycle check point) and tumour suppressor p53 to regulate the proliferation of senescent or unwanted cells [71,72]. In this study, a higher number of older donor BM-MSCs were positive for *β*-gal, indicating cellular senescence. These results were consistent with studies reporting a higher number of *β*-gal-positive cells in BM-MSCs from older donors [20,73]. The PD time assay, CFU-F assay, cell morphology study and SA-*β*-gal assay collectively demonstrated the reduced proliferation capacity of older donor BM-MSCs with features of senescence when cultured in standard conditions. Conversely, in the presence of SNC, the proliferation capacity of older donor BM-MSCs was similar to that of young donor BM-MSCs. In relation to the effects of SNC on senescent cells, we did not explore this directly, but we noticed that the attachment and growth of older donor BM-MSCs on SNC were comparable to young donor BM-MSCs, unlike their behaviour in 2D conditions. The qPCR results demonstrated similar levels of p16, p21 and p53 transcripts [74,75] in older and young donor MSCs grown on SNC after 7 days of osteoinduction. 

Several studies have reported a reduction in the osteogenic potential of older donor BM-MSCs. Zhou et al. demonstrated the reduced osteogenesis of human BM-MSCs in osteo-differentiation medium (14 days) with increasing donor age (17 to 90 years) [20]. Similarly, Chen et al. observed reduction in the osteogenesis of BM-MSCs with advancing age (15 to 85 years) after 12 days of osteoinduction [73]. Stolzing et al. and Carvalha et al. observed the same after 10 days of osteoinduction in BM-MSCs from people aged 15 to 55 years [45] and after 21 days of osteoinduction in BM-MSCs from people aged 60 to 80 years [66]. In the present study, both the young and older donor BM-MSCs on SNC were subjected to SM medium for 7 days (favors attachment) and then to osteo-differentiation medium for 14 days. SEM-EDS and q-PCR were used to study the osteogenesis of BM-MSCs cultured on SNC. The presence of native minerals in SNC limits the use of Alizarin Red or von Kossa staining for the detection of Ca/P deposited on SNC by BM-MSCs. In contrast, SEM-EDS mapping was performed to a depth of 2 µm, thus allowing the SNC sample to be distinguished from calcium/phosphorus signals produced by BM-MSCs [55]. Again, to specifically find the calcium and phosphorus deposited by BM-MSCs, the mass (%) values of the calcium and phosphorus present on the SNC with BM-MSCs were normalised with the average mass (%) values of the calcium and phosphorus present on the SNC alone. During SEM-EDS imaging, a visible mineralization matrix was observed on both the SNCs with young and older donor BM-MSCs, with evidence of overlapping calcium and phosphorus deposition. The amount of normalised mass (%) of calcium and phosphorus and the relative expression of osteogenesis-related genes in older donor BM-MSCs cultured on SNC were comparable to that of young donor BM-MSCs cultured on SNC. 

BMP2 plays a significant role in the commitment of MSCs to osteo-chondroprogenitor cells. Subsequently, the osteo-chondroprogenitor cells undergo commitment towards osteogenesis via the activation of RUNX2. These committed osteoprogenitor cells then differentiate into pre-osteoblasts with the expression of collagen I, ALP and SPARC [76]. OMD is a marker of osteoblast differentiation [77] and is a positive controller of osteogenesis through the BMP2/SMAD pathway [78]. SPARC is a non-collagenous protein secreted by osteoblasts and involved in collagen formation and mineralization [79,80]. We investigated the expression of BMP2, RUNX2 and ALP on day 14 of osteoinduction and observed the high-level expression of the mature osteoblast markers SPARC, COL1A1 and OMD, with no marked difference between young and older donor BM-MSCs. Altogether, the SEM-EDS and gene expression data suggested that SNC contributed to the osteogenesis of older donor BM-MSCs, which was comparable to that of young donor MSCs.

One major component of SNC is shell nacre (72 wt.%), which has already been shown to be osteogenic and angiogenic [81,82,83]. Earlier studies by Lopez et al. demonstrated the osteogenic potential of nacre chips *in vitro* without any chemical inducers [39,84]. Later, Green et al. demonstrated the osteogenic potential of shell nacre chips and the soluble matrix protein using human BM-MSCs [85]. Many *in vivo* studies have demonstrated the osteogenic potential and biocompatibility of shell nacre as a direct bone substitute [86,87,88,89,90,91,92,93,94]. Shell nacre containing hydrogel [95] and shell nacre containing calcium phosphate cement [58] showed enhanced osteogenic potential due to the addition of shell nacre. The water-soluble matrix (WSM) of shell nacre induced osteo-differentiation similar to that of BMP2 [96], dexamethasone [97] and rh-BMP2 [85]. Low-molecular-weight molecules [98,99], proteins of WSM [100,101], and calcium [102,103] ions of shell nacre can act as an inducerons [102,104,105,106], which play the role of signalling molecules to promote osteogenesis. Kim et al. demonstrated that WSM induced osteoblast mineralization through the c-jun NH2 terminal kinase and Fos-related antigen-1 (Fra-1) pathway [40]. In the current study, it is possible to suggest that the shell nacre component of SNC aided in the osteogenesis of older donor BM-MSCs to a level comparable to that of the young donor BM-MSCs. 

Although the older donor BM-MSCs exhibited a low proliferation capacity and contained a higher proportion of senescent cells, the presence of SNC aided in their attachment and growth to a level comparable to that of young donor BM-MSCs. The WSM of shell nacre promoted the cell proliferation of rat BM-MSCs [97] and prolonged the survival of osteoblasts due to the enhanced expression of anti-apoptotic protein Bcl-2 [96,107]. We believe that SNC may not directly reverse senescence, but that the presence of bioactive factors in shell nacre may have some anti-oxidant potential [108,109], which can slow down the senescence progression of older donor BM-MSCs. A previous study on old human-muscle-derived stem cells (hMDSCs) showed low levels of p38MAPK compared to young hMDSCs. This difference contributed to higher rates of survival of old hMDSCs and bone regeneration comparable to that of young hMDSCs [74]. Further studies involving Bcl-2 and p38 MAPK will reveal the mechanisms at play. 

As studied previously [110,111,112], local host cells including progenitors from the neighbouring muscle tissue can participate in bone repair; in the future, it would be very useful to study the interaction of SNC with muscle progenitor cells [74], as well as with MSCs from other surrounding tissues including adipose tissue [113]. Further work is also needed to evaluate the effect of the aging host systemic microenvironment on the bone repair at the implantation [74]. Numerous studies have explored gender-related differences in the proliferation and osteogenesis of BM-MSCs from humans [63,114]. However, conflicting findings exist, as some studies have not observed such differences in humans [11,115]. 

SNC was found to be non-cytotoxic to human BM-MSCs. SNC was conducive to their proliferation and supported osteogenic differentiation regardless of the donor’s age and the senescence status. One limitation of this study is the relatively short cultivation time used in the SM medium (7 days) when studying the proliferation of BM-MSCs on the SNC and the use of a potent osteo-differentiation medium for the comparison of the osteogenic potential of BM-MSCs on SNC. The use of medium without osteogenic supplements like dexamethasone and extending the duration of the osteogenesis study to up to 28 days would potentially reveal the full extent of spontaneous osteogenesis occurrence on SNC. We are also mindful that a larger cohort of donors would be required for future the investigation of this material. Considering gender differences, a further investigation with male and female donors would establish the versatility of SNC in both genders. Furthermore, an *in vivo* osteogenesis study in osteoporotic models may provide evidence for the superiority of SNC. 

In conclusion, to the best of our knowledge, this is the first study to investigate the osteogenic potential of SNC as a material by using BM-MSCs from young and older donors, where senescence is well recognised. This study underscores the need to evaluate the osteogenesis of any bone substitute material with BM-MSCs from older donors, considering the higher rates of osteoporosis-related orthopaedic surgery performed in the geriatric population.

## Figures and Tables

**Figure 1 bioengineering-11-00143-f001:**
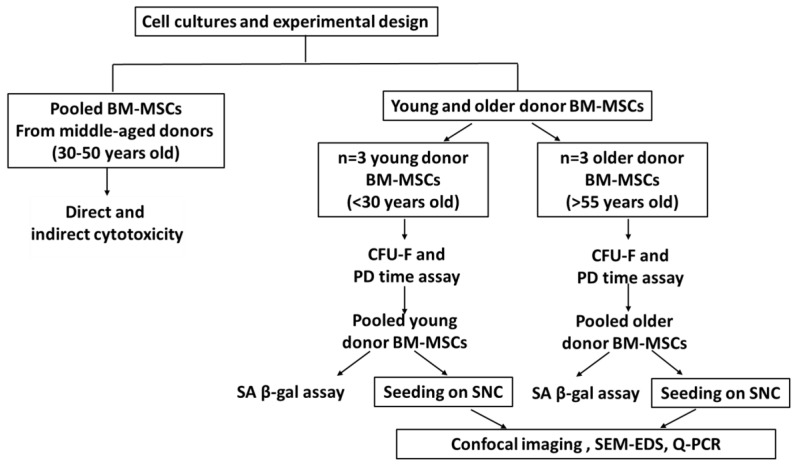
Flowchart of experimental design.

**Figure 2 bioengineering-11-00143-f002:**
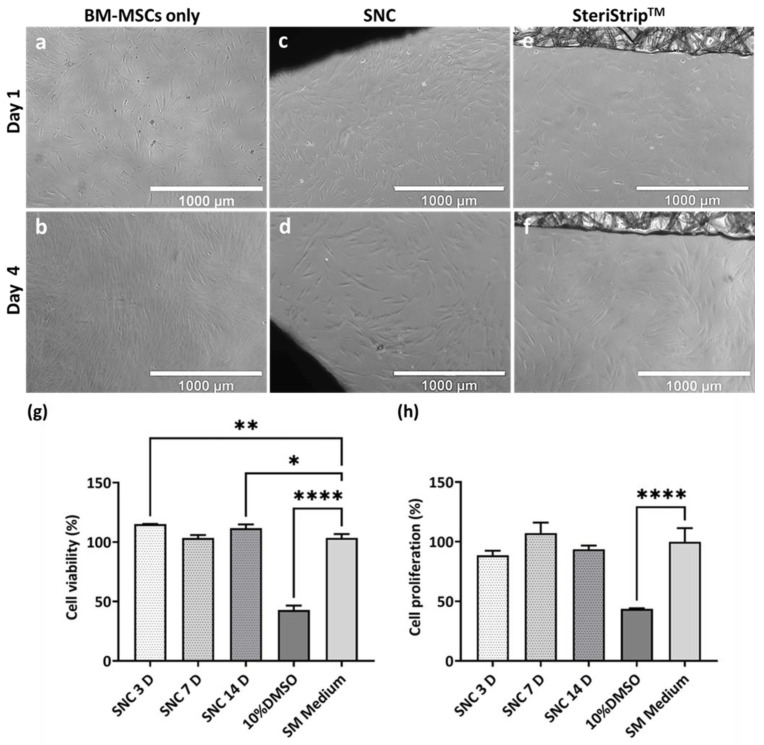
Direct contact and indirect cytotoxicity testing of SNC samples and extracts using BM-MSCs. (**a**–**f**) Light microscopy images of direct contact of SNC samples and SteriStrip^TM^ with BM-MSCs for 1 and 4 days. The scale bars represent 1000 µm. (**g**) Cell viability (%) of BM-MSCs after 24 h contact with SNC extracts and controls (**h**) Cell proliferation (%) of BM-MSCs after 4 days contact with SNC extracts and controls. Ordinary one-way ANOVA–Tukey’s multiple comparison test. *—*p* value < 0.05, **—*p* value < 0.01, ****—*p* value < 0.0001. Data shown as the mean ± standard deviation.

**Figure 3 bioengineering-11-00143-f003:**
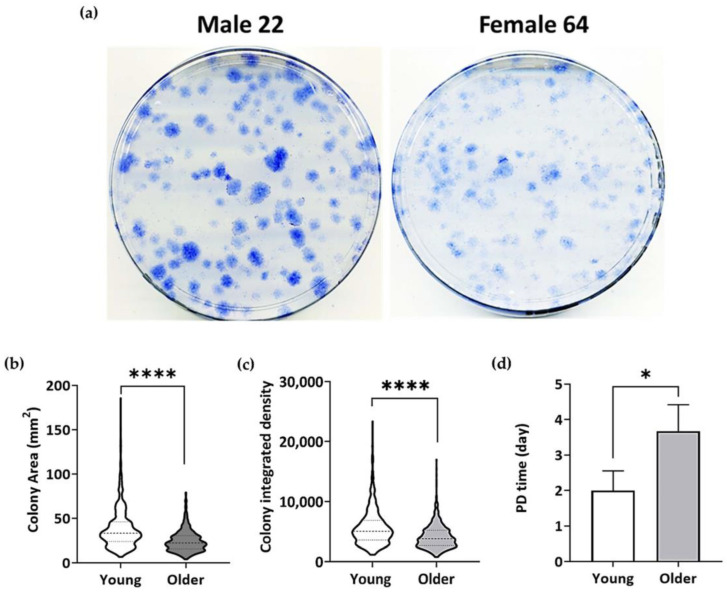
Comparison of young and older donor BM-MSCs. (**a**) Representative images of methylene-blue-stained CFU-F plates. (**b**) Violin plots showing the colony area distribution and (**c**) colony-integrated density of the BM-MSCs of young (*n* = 3) and older donors (*n* = 3). Mann–Whitney test ****—*p* < 0.0001. Data shown as violin plots with median (dashed lines) and interquartile range (dotted lines) values of colony area and colony-integrated density. (**d**) PD time of young and older donor BM-MSCs. Unpaired *t*-test, two-tailed, *—*p* < 0.05. Data shown as mean ± standard deviation.

**Figure 4 bioengineering-11-00143-f004:**
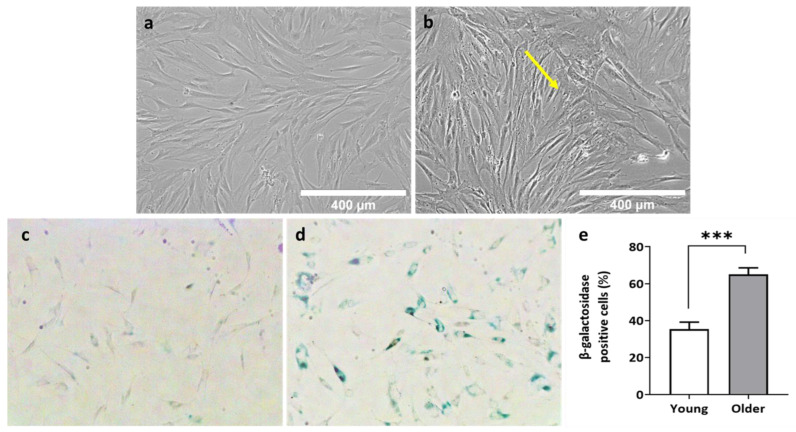
Cell morphology and SA-*β* Gal histochemical staining assay (**a**) Microscopic images of pooled young donor BM-MSCs and (**b**) older donor BM-MSCs grown in 3 days. Scale bars represent 400 µm and yellow arrow refers to senescent cell morphology. SA-*β* Gal histochemical staining images of (**c**) pooled young donor BM-MSCs and (**d**) pooled young donor BM-MSCs. Magnification 100X. (**e**) Senescence levels shown as the percentage of *β*-galactosidase-positive cells in pooled young donor BM-MSCs and in pooled older donor BM-MSCs using the SA-*β* Gal assay. Unpaired *t*-test. Two-tailed, ***—*p* < 0.001. Data shown as the mean ± standard deviation.

**Figure 5 bioengineering-11-00143-f005:**
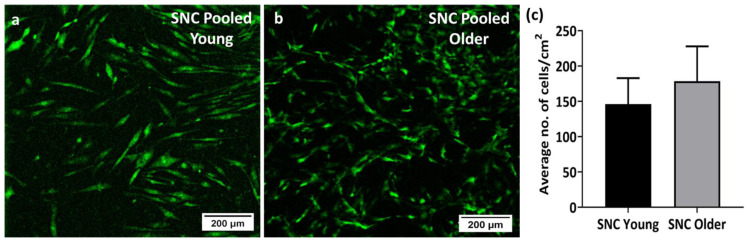
CellTracker^TM^ Green staining of BM-MSCs on SNC samples (day 7). Confocal microscopy images of (**a**) pooled young donor BM-MSCs and (**b**) pooled older donor BM-MSCs on SNC. (**c**) The average number of pooled young donor BM-MSCs and pooled older donor BM-MSCs present on SNC was counted using Image J v1.52 (*n* = 3 images) and the differences were not statistically significant. The scale bar represents 200 µm.

**Figure 6 bioengineering-11-00143-f006:**
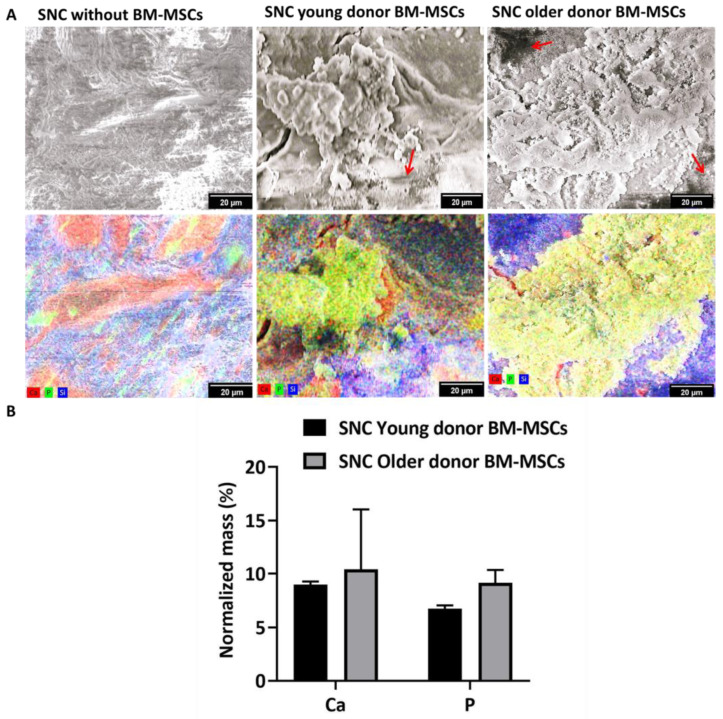
*In vitro* mineralization of BM-MSCs seeded on SNC: (**A**) Representative SEM-EDS images of SNC without cells, SNC with pooled young donor BM-MSCs and SNC with pooled older donor BM-MSCs. The red arrow indicates the presence of cells. Each element measured was superimposed onto the SEM image using pseudo colour, calcium (red), phosphorous (green) or silicon (blue). The scale bar represents 20 µm; (**B**) Semi-quantitative analysis of the normalised mass (%) of calcium and phosphorus (*n* = 3 images from different replicates). Data shown as mean ± standard deviation.

**Figure 7 bioengineering-11-00143-f007:**
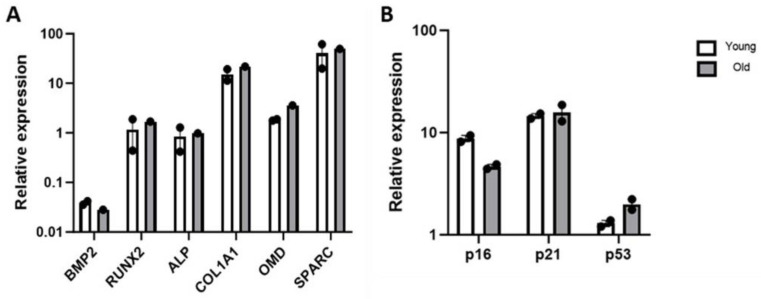
Relative expression of genes in young and older donor BM-MSCs cultured on SNC: (**A**) osteogenesis-related genes such as BMP2, RUNX2, ALP, COL1A1, OMD and SPARC (day 14 of osteoinduction) and (**B**) senescence/cell cycle-related genes such as p16, p21 and p53 (day 7 of osteoinduction). Error bars represent the median with the range.

## Data Availability

The data used to support the findings of this study are available from the corresponding author upon reasonable request.

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
