# Peer review of "In Vitro Osteogenesis Study of Shell Nacre Cement with Older and Young Donor Bone Marrow Mesenchymal Stem/Stromal Cells"

_bioengineering, 2024, doi:10.3390/bioengineering11020143_

Round 1

Reviewer 1 Report

Comments and Suggestions for Authors

In this study, authors investigated the in vitro osteogenic potential of SNC with iliac crest BM-MSCs from older and young donors. The topic is interesting. There are some concerns regarding experimental design and results. I would like to recommend a major revision before acceptance.

Specific comments are as follows:

1.     Figure 2, which group was taken as control for comparison? From microscopy, the proliferation of BM-MSC is much higher than other two cohorts. Is this consistent with the results below?

2.     Is only one case from young and older donor BM-MSCs sufficient for drawing the  conclusion? Please comment on this issue.

3.     Figure 4, which SA-β Gal assay was used in this work? Please provide the kit information in the method session and representative images for assay in Figure 4.

4.     In vitro osteogenic potential should be further confirmed by osteogenic marker tests, such as ALP. Related osteogenic gene analysis are encouraged. Only mineralization is not sufficient to support the conclusion. 

Comments on the Quality of English Language

Minor editing of English language required

Author Response

  1. Figure 2, which group was taken as control for comparison? From microscopy, the proliferation of BM-MSC is much higher than other two cohorts. Is this consistent with the results below?

Reply:

We thank the reviewer for their insightful comments. In response, we firstly clarified that the cytotoxicity experiments were performed with BM-MSCs from middle-aged donors (lines 138, 146). This was done to preserve precious BM-MSCs from older donors to be used in all the subsequent experiments. The comparisons in direct cytotoxicity assays were performed between SNC samples secured with SteriStripTM, SteriStripTM alone and cells alone. Indeed, the cell density in BM-MSCs only appears slightly higher than in the presence of SteriStripTM (with or without the SNC), which could reflect the effect of the SteriStripTM themselves. There were no visual differences between SteriStripTM alone and SteriStripTM +SNC confirming the non-toxic nature of the SNC. We have clarified this in the manuscript text, lines 403-408.

In indirect assays, there was no requirement for SteriStripTM to secure the scaffolds in place, instead the SNC extracts were used and the controls included the extraction media alone (SM media) and media containing DMSO (toxic to cells). Indirect assays showed no inhibition of BM-MSC proliferation with the addition of the 3-day SNC extracts, as well as longer-time extracts (7 days and 14 days). Therefore, indirect assays confirmed and extended our findings obtained using direct assays, this is now clarified in lines 407-412.

  1. Is only one case from young and older donor BM-MSCs sufficient for drawing the conclusion? Please comment on this issue.

Reply: Thank you so much for your pertinent observation. Many studies have used large patient cohorts and reported that increasing age reduces the proliferation and osteogenesis of older donor BM-MSCs (Churchman et al., 2017; El-Jawhari et al., 2021; Ganguly et al., 2019; Siegel et al., 2013; Stolzing et al., 2008; Vogt et al., 2023). As well, these and other studies have reported that donor variation within the same age group does exists and can be attributed to donor’s ’biological age’ as opposed to their ’chronological age’. For this reason, in the present study, we did not use single cases of young and older donor MSCs. Based on the previous experience with BM-MSC proliferation and senescence assays (Churchman et al., 2017; Zhai et al., 2021), the young and older donor groups consisted of n=3 donors each and they were at least 3 decades apart (Young less than 30 years and Older above 55 years). Indeed, using these group sizes we were able to confirm significant differences in cell proliferation and senescence. (Figures 3 and 4). For the SNC experiments, we were limited with the number of SNC scaffolds that could be manufactured and used in all the planned experiments. Therefore, for these experiments, we separately ‘pooled’ (combined) BM-MSCs from the same young donors (n=3) and older donors (n=3). Not only did this reduce the number of the required scaffolds, but it also minimized donor variation. This has now been highlighted in Discussion, lines 422-431, 437-438.

  1. Figure 4, which SA-β Gal assay was used in this work? Please provide the kit information in the method session and representative images for assay in Figure 4.

Reply: Thank you so much for the comment. We apologize for not including the name of the kit. The kit used is Senescence cell histochemical staining kit and the missing information is now added in Line 215. Based on the reviewer’s suggestion, the representative images of the senescence assay are now included in Figure 4 as figure 4c, 4d.

  1. In vitro osteogenic potential should be further confirmed by osteogenic marker tests, such as ALP. Related osteogenic gene analysis are encouraged. Only mineralization is not sufficient to support the conclusion. 

Reply: Thank you so much for the comment. We have now included the gene expression of bone morphogenetic protein 2 (BMP2), RUNT related transcription factor (RUNX2), alkaline phosphatase (ALP), Collagen type I Alpha 1 chain (COL1A1), Osteomodulin (OMD), Secreted protein acidic and cysteine rich (SPARC) as a separate figure 7. The corresponding methods line 255-272, results (line 374-381) and discussion are added to lines,477-493. The corresponding Taqman probes are now presented in the Supplementary table 1.

References

Churchman, S.M., Boxall, S.A., McGonagle, D., Jones, E.A., 2017. Predicting the Remaining Lifespan and Cultivation-Related Loss of Osteogenic Capacity of Bone Marrow Multipotential Stromal Cells Applicable across a Broad Donor Age Range. Stem Cells Int 2017, 6129596. https://doi.org/10.1155/2017/6129596

El-Jawhari, J.J., Ganguly, P., Jones, E., Giannoudis, P.V., 2021. Bone Marrow Multipotent Mesenchymal Stromal Cells as Autologous Therapy for Osteonecrosis: Effects of Age and Underlying Causes. Bioengineering 8, 69. https://doi.org/10.3390/bioengineering8050069

Ganguly, P., El-Jawhari, J.J., Burska, A.N., Ponchel, F., Giannoudis, P.V., Jones, E.A., 2019. The Analysis of In Vivo Aging in Human Bone Marrow Mesenchymal Stromal Cells Using Colony-Forming Unit-Fibroblast Assay and the CD45lowCD271+ Phenotype. Stem Cells International 2019, e5197983. https://doi.org/10.1155/2019/5197983

Siegel, G., Kluba, T., Hermanutz-Klein, U., Bieback, K., Northoff, H., Schäfer, R., 2013. Phenotype, donor age and gender affect function of human bone marrow-derived mesenchymal stromal cells. BMC Medicine 11, 146. https://doi.org/10.1186/1741-7015-11-146

Stolzing, A., Jones, E., McGonagle, D., Scutt, A., 2008. Age-related changes in human bone marrow-derived mesenchymal stem cells: consequences for cell therapies. Mech Ageing Dev 129, 163–173. https://doi.org/10.1016/j.mad.2007.12.002

Vogt, A., Kapetanos, K., Christodoulou, N., Asimakopoulos, D., Birch, M.A., McCaskie, A.W., Khan, W., 2023. The Effects of Chronological Age on the Chondrogenic Potential of Mesenchymal Stromal Cells: A Systematic Review. International Journal of Molecular Sciences 24, 15494. https://doi.org/10.3390/ijms242015494

Zhai, W., Tan, J., Russell, T., Chen, S., McGonagle, D., Win Naing, M., Yong, D., Jones, E., 2021. Multi-pronged approach to human mesenchymal stromal cells senescence quantification with a focus on label-free methods. Sci Rep 11, 1054. https://doi.org/10.1038/s41598-020-79831-9

Reviewer 2 Report

Comments and Suggestions for Authors

Peer-review bioengineering – 2752967

The research manuscript entitled “In Vitro Osteogenesis Study of Shell Nacre Cement with Older and Young Donor Bone Marrow Mesenchymal Stem Cells” describes an interesting and well-organized study. The manuscript topic is within the scope of the journal Bioengineering (ISSN 2306-5354) (section Regenerative Engineering) and of the special issue “Inspired by Nature: Advanced Biomaterials and Manufacturing Solutions for Skeletal Tissue Regeneration and Osteoarthritis Treatment”. However, to support the claims regarding the osteogenesis of BM-MSCs, the manuscript needs additional experiments and major improvements before being considered for publication.

Comments/Suggestions:

1.     The authors should follow the latest trends in the scientific community and define MSCs as mesenchymal stem/stromal cells.

2.     The authors should clarify and correct reference 44, which reports data on the development and physicochemical characterization of Shell Nacre Cement. Is this from a journal?

3.     (Materials and methods, subsection 2.2, lines 114, 115): Please clarify the number of BMA donors (n=10 or n=6?).

4.     (Materials and methods, subsection 2.4): The authors mentioned that for the indirect cytotoxicity assays, the media extractions were stored at -80ºC. Why the assay was not performed with freshly obtained extracts? Could this storage produce an effect on the results obtained?

5.     (Results, Section 3.3): The images of the SA-β Gal staining must be added to the manuscript or at least provided as supplementary material.

6.     (Results, Section 3.4, line 317): The differentiation was done for 14 or 21 days? Please clarify.

7.     The results provided are insufficient to support the authors' claim that the SNC supports BM-MSCs proliferation and in vitro osteogenesis. The proliferation of BM-MSCs on the SNC scaffolds should be monitored over time using cell metabolic activity assays (e.g. Alamar Blue or MTT) or DNA content quantification. Also, the osteogenic differentiation of BM-MSCs on the SNS scaffolds should be assessed by quantifying Alkaline phosphatase (ALP) activity, analyzing the expression of typical bone gene markers (RT-qPCR analysis) and of typical bone ECM proteins (collagen type I, osteopontin, etc) by immunofluorescence analysis. These experiments are quite important to support the authors' claim “SNC was conducive to their proliferation and supported osteogenic differentiation regardless the donor age or BM-MSC and senescence status.”

Minor issues:

1.     (Materials and methods, subsection 2.4, line 140): Please correct the verb form: “SNC samples (diameter 4mm x 1mm) were...”

2.     (Materials and methods, subsection 2.5.2, line 180): Please add the missing words “of the experiment” to obtain: “After the duration of the experiment, (...)”

3.     (Materials and methods, subsection 2.5.2, lines 182/183): Please specify the duration of the methylene blue staining protocol.

4.     (Materials and methods, subsection 2.6.2, line 206): Please specify the duration of the X-Gal staining mixture. What is the composition of the mixture?

5.     (Materials and methods, subsection 2.6.4 title, line 224): Please remove the “extra” point in the section title.

6.     (Materials and methods, subsection 2.7 title, line 243): Please correct the subsection numbering to “2.7 Statistical Analysis”.

7.     (Materials and methods, subsection 2.7, line 248): Please use the capital letter in “The symbols used (...)”

8.     (lines 381-383): The sentence ”While older donor BM-MSCs exhibited senescence features and a low proliferation capacity plated on tissue culture plastic.” Needs to be completed.

Comments on the Quality of English Language

Peer-review bioengineering – 2752967

The research manuscript entitled “In Vitro Osteogenesis Study of Shell Nacre Cement with Older and Young Donor Bone Marrow Mesenchymal Stem Cells” describes an interesting and well-organized study. The manuscript topic is within the scope of the journal Bioengineering (ISSN 2306-5354) (section Regenerative Engineering) and of the special issue “Inspired by Nature: Advanced Biomaterials and Manufacturing Solutions for Skeletal Tissue Regeneration and Osteoarthritis Treatment”. However, to support the claims regarding the osteogenesis of BM-MSCs, the manuscript needs additional experiments and major improvements before being considered for publication.

Comments/Suggestions:

1.     The authors should follow the latest trends in the scientific community and define MSCs as mesenchymal stem/stromal cells.

2.     The authors should clarify and correct reference 44, which reports data on the development and physicochemical characterization of Shell Nacre Cement. Is this from a journal?

3.     (Materials and methods, subsection 2.2, lines 114, 115): Please clarify the number of BMA donors (n=10 or n=6?).

4.     (Materials and methods, subsection 2.4): The authors mentioned that for the indirect cytotoxicity assays, the media extractions were stored at -80ºC. Why the assay was not performed with freshly obtained extracts? Could this storage produce an effect on the results obtained?

5.     (Results, Section 3.3): The images of the SA-β Gal staining must be added to the manuscript or at least provided as supplementary material.

6.     (Results, Section 3.4, line 317): The differentiation was done for 14 or 21 days? Please clarify.

7.     The results provided are insufficient to support the authors' claim that the SNC supports BM-MSCs proliferation and in vitro osteogenesis. The proliferation of BM-MSCs on the SNC scaffolds should be monitored over time using cell metabolic activity assays (e.g. Alamar Blue or MTT) or DNA content quantification. Also, the osteogenic differentiation of BM-MSCs on the SNS scaffolds should be assessed by quantifying Alkaline phosphatase (ALP) activity, analyzing the expression of typical bone gene markers (RT-qPCR analysis) and of typical bone ECM proteins (collagen type I, osteopontin, etc) by immunofluorescence analysis. These experiments are quite important to support the authors' claim “SNC was conducive to their proliferation and supported osteogenic differentiation regardless the donor age or BM-MSC and senescence status.”

Minor issues:

1.     (Materials and methods, subsection 2.4, line 140): Please correct the verb form: “SNC samples (diameter 4mm x 1mm) were...”

2.     (Materials and methods, subsection 2.5.2, line 180): Please add the missing words “of the experiment” to obtain: “After the duration of the experiment, (...)”

3.     (Materials and methods, subsection 2.5.2, lines 182/183): Please specify the duration of the methylene blue staining protocol.

4.     (Materials and methods, subsection 2.6.2, line 206): Please specify the duration of the X-Gal staining mixture. What is the composition of the mixture?

5.     (Materials and methods, subsection 2.6.4 title, line 224): Please remove the “extra” point in the section title.

6.     (Materials and methods, subsection 2.7 title, line 243): Please correct the subsection numbering to “2.7 Statistical Analysis”.

7.     (Materials and methods, subsection 2.7, line 248): Please use the capital letter in “The symbols used (...)”

8.     (lines 381-383): The sentence ”While older donor BM-MSCs exhibited senescence features and a low proliferation capacity plated on tissue culture plastic.” Needs to be completed.

Author Response

  1. The authors should follow the latest trends in the scientific community and define MSCs as mesenchymal stem/stromal/stromal cells.

Reply: Thank you so much for the comment. It is now changed in the whole document as bone marrow mesenchymal stem/ stromal cells.

  1. The authors should clarify and correct reference 44, which reports data on the development and physicochemical characterization of Shell Nacre Cement. Is this from a journal?

Reply: Thank you very much for the comment. We apologize for the mistake and it is corrected as (Reference 44. Line 681-683)

Wilson, B.J.; Philipose Pampadykandathil, L. Novel Bone Void Filling Cement Compositions Based on Shell Nacre and Siloxane Methacrylate Resin: Development and Characterization. Bioengineering 2023, 10, 752, doi:10.3390/bioengineering10070752.

  1. (Materials and methods, subsection 2.2, lines 114, 115): Please clarify the number of BMA donors (n=10 or n=6?).

Reply: Thank you so much for the comment. The number of BMA donors is 10. It is corrected as following (Line 121-126)

Out of these 10 donors, n=3 were categorized as ‘young’ donors aged below 30 (ranging between 21-26 years old, 3 males), n=3 were classified as ‘older’ donors aged above 55 years old (between 58-64 years old, 2 males, 1 female) and n=4 were classified as ‘middle-aged donors’ with age ranging from 30 to 50 years old (2 males, 2 females).

  1. (Materials and methods, subsection 2.4): The authors mentioned that for the indirect cytotoxicity assays, the media extractions were stored at -80ºC. Why the assay was not performed with freshly obtained extracts? Could this storage produce an effect on the results obtained?

Reply: Thank you so much for the insightful comment. Indeed, media extractions were performed for 3, 7 and 14 days, after which the samples were stored at −80°C until experiments with BM-MSCs were performed. This was done for two reasons: 1. to maintain consistency across all time points with the use of the same stock/passage of BM-MSCs and 2. to reduce variability by performing experiments on extracts from all time points together in the same 96-well plate Furthermore, our experimental set-up has maximised on the use of 96-well plates and reagents needed for the XTT assay. We acknowledge that storage in −80° C might have affected the integrity of some eluted chemicals, but we deem this unlikely given the chemical composition of SNC (Wilson and Philipose Pampadykandathil, 2023), and based on our previously published method for the same method (Iqbal et al., 2022; Yildizbakan et al., 2023)

  1. (Results, Section 3.3): The images of the SA-β Gal staining must be added to the manuscript or at least provided as supplementary material.

Reply: Thank you so much for the comment. It is now added as Figure 4c, d.

  1. (Results, Section 3.4, line 317): The differentiation was done for 14 or 21 days? Please clarify.

Reply: Thank you so much for the query. The osteo-differentiation was carried out for a maximum of 14 days (Line 240-243, 356-358); all the time points have been checked and if necessary, corrected throughout the whole document

  1. The results provided are insufficient to support the authors' claim that the SNC supports BM-MSCs proliferation and in vitro osteogenesis. The proliferation of BM-MSCs on the SNC scaffolds should be monitored over time using cell metabolic activity assays (e.g. Alamar Blue or MTT) or DNA content quantification. Also, the osteogenic differentiation of BM-MSCs on the SNS scaffolds should be assessed by quantifying Alkaline phosphatase (ALP) activity, analyzing the expression of typical bone gene markers (RT-qPCR analysis) and of typical bone ECM proteins (collagen type I, osteopontin, etc) by immunofluorescence analysis. These experiments are quite important to support the authors' claim “SNC was conducive to their proliferation and supported osteogenic differentiation regardless the donor age or BM-MSC and senescence status.”

Reply: Thank you so much for the valuable comments. We acknowledge the importance of comprehensive experiments to support our claim regarding SNC’s support for the proliferation and osteogenesis of BM-MSCs from both young and older donors. However, we were limited by the numbers of scaffolds and the limited availability and low proliferation capacity of older donor BM-MSCs. We have consented 14 older donors who have provided variable amounts of BM aspirates for expansion. Out of these, only 3 yielded sufficient cells needed for all the experiments. This is not uncommon given our previous experience with older donor BM-MSCs (Churchman et al., 2017). We therefore had to choose between different end-point assays and have opted for SEM-EDS mineralisation assay for the functional assessment of BM-MSC osteogenesis. We have previously used this assay in conjunction with ALP enzymatic activity assay, and the results were complimentary (Kouroupis et al., 2013).

In response to this and other reviewer comments, we have now included q-PCR results to further support our findings in respect to osteogenesis (please refer to our new Figure 7, The corresponding methods line 255-272, results (line 374-381) and discussion are added to lines,477-493. The corresponding Taqman probes are now presented in the Supplementary table 1. We appreciate your feedback and believe these adjustments have strengthened the robustness of the study.

Minor issues:

  1. (Materials and methods, subsection 2.4, line 140): Please correct the verb form: “SNC samples(diameter 4mm x 1mm) were...”

Reply: Thank you so much. It is corrected. Line 137-138.

  1. (Materials and methods, subsection 2.5.2, line 180): Please add the missing words “of the experiment” to obtain: “After the duration of the experiment, (...)”

Reply: Thank you so much. It is modified as follows

Line 191,192: After 2 weeks of culture, the media were removed, dishes washed in phosphate buffer saline (PBS, Life Technologies, Paisley, UK)

  1. (Materials and methods, subsection 2.5.2, lines 182/183): Please specify the duration of the methylene blue staining protocol.

Reply: Thank you so much. The sentence is modified as follows

Line 192-195: After 2 weeks of culture, the media were removed, dishes washed in phosphate buffer saline (PBS, Life Technologies, Paisley, UK), the cells fixed with 3.7% formaldehyde (Fisher Scientific, Loughborough, UK), and stained with 1% w/v methylene blue (Sigma, Dorset, UK) for 30 mins.

  1. (Materials and methods, subsection 2.6.2, line 206): Please specify the duration of the X-Gal staining mixture. What is the composition of the mixture?

Reply: Thank you so much for the query. The requested detail is now added

Line 215-220. Following a PBS wash, the cells were stained with X-Gal staining mixture (1 ml) prepared as per the manufacturer's protocol (Senescence Cells histochemical staining kit, Sigma Aldrich, USA). In brief, 10 ml of the staining mixture included 400 mM Potassium Ferricyanide (125 µl), 400 mM Potassium Ferrocyanide (125 µl), X-gal solution (40 mg/ml, 1 ml) and ultrapure water (8.5 ml). The dishes were sealed in parafilm and incubated at 37°C without CO2 overnight.

  1. (Materials and methods, subsection 2.6.4 title, line 224): Please remove the “extra” point in the section title.

Reply: Thank you so much. It was corrected as following

“In vitro osteogenic potential of BM-MSCs cultured on SNC” Line 235

  1. (Materials and methods, subsection 2.7 title, line 243): Please correct the subsection numbering to “2.7 Statistical Analysis”.

Reply: Thank you so much. It was corrected as 2.7 Line 276

  1. (Materials and methods, subsection 2.7, line 248): Please use the capital letter in “The symbols used (...)”

Reply: Thank you so much. It is now corrected Line 282

  1. (lines 381-383): The sentence” While older donor BM-MSCs exhibited senescence features and a low proliferation capacity plated on tissue culture plastic.” Needs to be completed.

Reply: Thank you so much. The sentence is modified as below (Line 448-452)

The PD time assay, CFU-F assay, cell morphology study and SA-gal assay collectively demonstrated the reduced proliferation capacity of older donor BM-MSCs with features of senescence when cultured in standard conditions. Conversely, in the presence of SNC, the proliferation capacity of older donor BM-MSCs was similar to that of young donor BM-MSCs.

References

Churchman, S.M., Boxall, S.A., McGonagle, D., Jones, E.A., 2017. Predicting the Remaining Lifespan and Cultivation-Related Loss of Osteogenic Capacity of Bone Marrow Multipotential Stromal Cells Applicable across a Broad Donor Age Range. Stem Cells Int 2017, 6129596. https://doi.org/10.1155/2017/6129596

Iqbal, N., Braxton, T.M., Anastasiou, A., Raif, E.M., Chung, C.K.Y., Kumar, S., Giannoudis, P.V., Jha, A., 2022. Dicalcium Phosphate Dihydrate Mineral Loaded Freeze-Dried Scaffolds for Potential Synthetic Bone Applications. Materials 15, 6245. https://doi.org/10.3390/ma15186245

Kouroupis, D., Baboolal, T.G., Jones, E., Giannoudis, P.V., 2013. Native multipotential stromal cell colonization and graft expander potential of a bovine natural bone scaffold. Journal of Orthopaedic Research 31, 1950–1958. https://doi.org/10.1002/jor.22438

Wilson, B.J., Philipose Pampadykandathil, L., 2023. Novel Bone Void Filling Cement Compositions Based on Shell Nacre and Siloxane Methacrylate Resin: Development and Characterization. Bioengineering 10, 752. https://doi.org/10.3390/bioengineering10070752

Yildizbakan, L., Iqbal, N., Ganguly, P., Kumi-Barimah, E., Do, T., Jones, E., Giannoudis, P.V., Jha, A., 2023. Fabrication and Characterisation of the Cytotoxic and Antibacterial Properties of Chitosan-Cerium Oxide Porous Scaffolds. Antibiotics 12, 1004. https://doi.org/10.3390/antibiotics12061004

Reviewer 3 Report

Comments and Suggestions for Authors

This in vitro study compared young and relative old bone marrow MSCs for their proliferation and survival on SNC. Overall, the study goal is clear and data are presented clearly. But lack of novelty and in vivo study and no mechanistic study.  Following are some of my comments for the authors to improve the manuscript.

Line 232, How does the author use SEM-EDS analysis to quantify mineralization elements such as calcium and phosphates? More details should be given. Why the Alizarin Red or Von Kossa staining was not used for detecting mineralization which is commonly used? Does SNC limit this detection? Is SNC absorbable? Since SNC is cement, does the material have calcium and Phosphate itself? How does the author know these are deposited by BM-MSCs?

Line 271, How does the author measure cell proliferation? XTT assay is color absorption? More details are needed. How cell proliferation is calculated needs more details.

Figure 4 : Authors should show beta-gal staining, 65% cells that become senescent are quite high.

Figure 5 legend, BM_MSC should be BM-MSC.

Discussion, previous study comparing young and old muscle-derived stem cells for bone regeneration should be cited (30463597) and the effect of age on MSCs isolated from different tissues also have been studied (25541697). How SNC can stimulate better osteogenic differentiation for old senescent cells? Can SNC reverse cellular senescence? Senescent cells are known to lose function and are detrimental to other cells. 

Comments on the Quality of English Language

English grammar is acceptable. 

Author Response

Line 232, How does the author use SEM-EDS analysis to quantify mineralization elements such as calcium and phosphates? More details should be given.

Reply: Thank you so much for the query. The detailed SEM-EDS is as follows (Line 242-253).

               At the end of the culture (day 14 of osteoinduction), mineral deposition was studied by SEM-EDS, as previously described (Kouroupis et al., 2013). The control group comprised of triplicate SNC samples without cells, similarly treated with the SM medium for 7 days and osteo-differentiation medium for 14 days. At the end of the culture, the samples were washed thrice with PBS and fixed with 3.7% paraformaldehyde. Following dehydration in increasing ethanol concentrations (30% to 100%), the samples were then evaluated using SEM-EDS. In brief, EDS mapping was performed to a depth of 2 µm to enable the measurements of calcium/phosphorus signals in the neighbourhood of cells and near the SNC surface. The spectrum with mass (%) values of calcium and phosphorus was obtained for each replicate sample and the values were normalized to the average mass (%) of calcium and phosphorus present in the control SNC samples and compared between young and older donor BM-MSCs.

Why the Alizarin Red or Von Kossa staining was not used for detecting mineralization which is commonly used? Does SNC limit this detection?

Reply: Thank you so much for the query. We acknowledge that Alizarin and von Kossa are commonly used staining for detecting calcium and phosphate mineralization, and we have previously employed Alizarin Red staining to detect mineralisation within collagen-contained scaffolds (El-Jawhari et al., 2019). The reviewer is absolutely right in suggesting that the presence of native minerals in shell nacre/SNC limits the use of these methods for the detection of Ca/P deposited on SNC by BM-MSCs. In contrast, SEM/EDS mapping that we have selected is performed to a depth of 2 µm on a dense cellular layer thus allowing the distinction of calcium/phosphorus signals from mineralizing cells and the SNC sample itself. This is included in discussion line 464-471.

Is SNC absorbable? Since SNC is cement, does the material have calcium and Phosphate itself? How does the author know these are deposited by BM-MSCs?

 Reply: Thank you so much for the query.  SNC is not absorbable and it inherently contains calcium. SNC is a chemically cured cement with compressive strength of 110 MPa (Wilson and Philipose Pampadykandathil, 2023)  As stated in line 243-245, SNC alone and SNC with BM-MSCs were subjected to same treatment medium. To specifically find the calcium and phosphorus deposited by BM-MSCs, the mass (%) values of calcium and phosphorus present on the SNC with BM-MSCs was normalized with the average mass (%) values of calcium and phosphorus present on SNC alone. The responses are now added to Discussion, lines 464 -474.

Line 271, How does the author measure cell proliferation? XTT assay is color absorption? More details are needed. How cell proliferation is calculated needs more details.

Reply: Thank you so much for the query.

XTT is a calorimetric assay. The yellow coloured XTT is reduced to orange coloured formazan in the presence of metabolically active cells indicative of the activity of mitochondrial dehydrogenase in the samples.  This change of colour can be directly correlated to the metabolic activity of the cells and thus can be quantified to calculate cell proliferation. The orange colour developed by the metabolically active cells is read at 450 nm using spectrophotometer. The percentage metabolic activity of the cells was calculated using the formula: (OD test/OD cell only SM Medium control) *100. The percentage of metabolically activity thus corresponds to the cell viability/proliferation. These details were updated in the XTT procedure given in Line 162-167.

Figure 4: Authors should show beta-gal staining, 65% cells that become senescent are quite high.

Reply: Thank you so much. Indeed, 65% cells that become senescent are quite high but this was expected given the average older donor age was 60 and BM-MSC passage number used was p2 to p3. These images from the senescence assay were requested by other reviewers and are now included as fig. 4c and 4d.

Figure 5 legend, BM_MSC should be BM-MSC.

Reply: Thank you so much.  BM-MSC is corrected in figure 5 legend.

Discussion, previous study comparing young and old muscle-derived stem cells for bone regeneration should be cited (30463597) and the effect of age on MSCs isolated from different tissues also have been studied (25541697). How SNC can stimulate better osteogenic differentiation for old senescent cells? Can SNC reverse cellular senescence? Senescent cells are known to lose function and are detrimental to other cells. 

Reply: Thank you so much the valuable comments. The suggested references are now incorporated in Discussion, lines 520-525. These new sentences read: As shown previously (Julien et al., 2021; Owston et al., 2016; Shah et al., 2013) local host cells including progenitors from the neighbouring muscle tissue can participate in bone repair and in the future, it would be very important to study the interaction of SNC with muscle progenitor cells (Gao et al., 2018) as well as MSC from other surrounding tissues including adipose tissue (Beane et al., 2014). Further work is also needed to evaluate the effect of the ageing host systemic microenvironment on the bone repair at the implantation (Gao et al., 2018).

In our original version we suggested that shell nacre could have been the main constituent of SNC responsible for improving the osteogenenic performance of older donor BM MSCs. We have re-written this paragraph, which now reads: (lines 492-507 in the revision): One major composition of SNC is shell nacre (72 wt.%), which has already been shown to be osteogenic and angiogenic (Gerhard et al., 2017; Willemin et al., 2019; Zhang et al., 2017). Earlier studies by Lopez et al demonstrated osteogenic potential of nacre chips in vitro without any chemical inducers (Lopez et al., 1992; Silve et al., 1992). Later, Green et al., demonstrated the osteogenic potential of shell nacre chips and soluble matrix protein using human BM-MSCs, (Green et al., 2015). Many in vivo studies have demonstrated the osteogenic potential and biocompatibility of shell nacre as a direct bone substitute (Atlan et al., 1999, 1997; Berland et al., 2005; Delattre et al., 1997; Iandolo et al., 2022; Lamghari et al., 2001b, 2001a, 1999; Leelatian et al., 2022). Shell nacre containing hydrogel (Flausse et al., 2013) and shell nacre containing calcium phosphate cement (Ruan et al., 2018) showed enhanced osteogenic potential due to the addition of shell nacre. Water soluble matrix (WSM) of shell nacre induced osteo-differentiation similar to that of BMP-2 (Pereira Mouriès et al., 2002), dexamethasone (Almeida et al., 2001) and rh-BMP-2 (Green et al., 2015). Low molecular weight molecules (Bédouet et al., 2006; Rousseau et al., 2008), proteins (Lao et al., 2007; Zhang et al., 2006) of WSM and calcium (Daneshmandi et al., 2022; Hosseini et al., 2021) ions of shell nacre can act as an inducerons (Cushnie et al., 2014; Daneshmandi et al., 2022; Laurencin and Nair, 2016; Ogueri et al., 2019) which play the role of signalling molecules to promote osteogenesis. Kim et al. demonstrated that WSM induced osteoblast mineralization through the c-jun NH2 terminal kinase and Fos-related antigen-1 (Fra-1) pathway (Kim et al., 2012). In the current study, it is possible to suggest that shell nacre component of SNC aided in the osteogenesis of older donor BM-MSCs to the level comparable to that of young donor BM-MSCs.

In relation to SNC effects on senescent cells, we did not explore this directly but we noticed that the attachment and growth of older- BM-MSCs on SNC was comparable to younger-donor BM-MSCs, unlike their behaviour in 2D conditions. We have now added qPCR results to this section of the study, which demonstrate similar levels of p16, p21 and p53 transcripts, often used to explore senescence (Gao et al., 2018; Zhai et al., 2019) in older- and younger donor MSCs (Supplementary Figure1) grown on SNC. We have also modified the following paragraph in Discussion, which now reads (lines 508-519):

            Although the older donor BM-MSCs exhibited low proliferation capacity and contained higher proportion of senescent cells, the presence of SNC aided in their improved attachment and growth comparable to that of young donor BM-MSCs. The WSM of shell nacre promoted cell proliferation of rat BM-MSCs (Almeida et al., 2001) and prolonged the survival of osteoblast with enhanced expression of anti-apoptotic protein Bcl-2 (Moutahir-belqasmi et al., 2001; Pereira Mouriès et al., 2002). We believe that SNC may not directly reverse senescence, but the presence of bioactive factors in shell nacre may have some anti-oxidant potential (Chaturvedi et al., 2013), which can slow down the senescence progression of older donor BM-MSCs. A previous study on old human muscle derived stem cells (hMDSCs) showed low levels of p38MAPK than young hMDSCs. This difference contributed to higher rates of survival of old hMDSCs and bone regeneration than young hMDSCs (Gao et al., 2018). Further studies involving Bcl-2 and p38 MAPK will reveal the mechanisms at play.

Reviewer 4 Report

Comments and Suggestions for Authors

Specific points and suggestions for improvement of the manuscript are listed below.

General comments:
(1) The number of individuals in total and their grouping is somewhat confusing. L114 lists 10 individuals (n=10) that were harvested for bone marrow. Then, in L115 the authors’ state: “out of these 6 donors…”. Next, in L121 the text reads: “Frozen vials of previously characterized BM-MSCs (age range 21-64 years old) were defrosted …”. Are these the missing 4 donors that will bring the count to n=10 (6+4)?

Then, in L129 the authors describe another middle-aged donor group (30 to 50 years old). Are these donors part of the previous 10 or in addition.

(2) In relation to the previous comment, why are the authors using a middle-aged donor group when their entire experiment uses young donors (n=3; age range of 21-26 years old), and old donors (n=3; age range between 58-64 years old). This middle-aged donor group was used for one purpose only - Direct Cytotoxicity Studies. Why not run this test on cells from the young and old groups?

(3) Please add the sex of the donors. Also – is there any evidence in the literature for a different behavior of BM-MSCs between males and females?

(4) Experiments were carried out for 72 h, 7 days and 14 days. Why not use 3, 7 and 14 days? What is the importance of 72 hours?

(5) Text on scale bar (figures 2 and 4) are too small to see. Please enlarge (see figures 5 and 6).

(6) Please always use the same significant decimal digits per parameters. For example: L288 “3.67 +/- 0.6” should be “3.7 +/- 0.6”; L290: “1.99 +/- 0.6” should be “2.0 +/- 0.6”.

(7) Violin plots in figure 3: please explain in the legend what are the sashed and dotted lines (median, interquartile range?).

(8) can the authors add an image of the “SNC samples without the cells” (L318) to Figure 6 as this is the control. This will allow the reader to compare the results visually.

(9) In several places in the manuscript the authors describe cell viability % above 100% (e.g., L261-270, L348). Can the authors explain shortly in the text what does it mean?

Specific comments:

- L42: “BM-MSCs”. This is the first time this abbreviation appears in the main text, please spell it out in full.

- L45: “BM”. This is the first time this abbreviation appears in the main text, please spell it out in full.

- L46: “bone modeling”. Suggesting to change to “bone modeling and remodeling”.

- L66: “resorb-ability” should be “resorbability”

- L103 “other ingredients” – please list them.

- L123: “SM”. This is the first time this abbreviation appears in the main text, please spell it out in full (Skim Milk?).

- L281: what is the unit ID (5050 +/- 88 ID)? Colony integrated density? Please explain that in the text.

- L289: ”p=0.0334” please change to P ≤ 0.05.

Author Response

General comments:
(1) The number of individuals in total and their grouping is somewhat confusing. L114 lists 10 individuals (n=10) that were harvested for bone marrow. Then, in L115 the authors’ state: “out of these 6 donors…”. Next, in L121 the text reads: “Frozen vials of previously characterized BM-MSCs (age range 21-64 years old) were defrosted …”. Are these the missing 4 donors that will bring the count to n=10 (6+4)? Then, in L129 the authors describe another middle-aged donor group (30 to 50 years old). Are these donors part of the previous 10 or in addition.

Reply: Thank you so much for the query. We apologize for the confusion created. The number of BMA donors is 10. It is corrected as following (Line 121-126)

BMA was collected in ethylenediaminetetraacetic acid (EDTA) tubes to prevent coagulation and harvested from n=10 donors between 21-64 years old. Out of these 10 donors, n=3 were categorized as ‘young’ donors aged below 30 (ranging between 21-26 years old, 3 males), n=3 were classified as ‘older’ donors aged above 55 years old (between 58-64 years old, 2 males, 1 female) and n=4 were classified as ‘middle-aged donors’ with age ranging from 30 to 50 years old (2 males, 2 females).

(2) In relation to the previous comment, why are the authors using a middle-aged donor group when their entire experiment uses young donors (n=3; age range of 21-26 years old), and old donors (n=3; age range between 58-64 years old). This middle-aged donor group was used for one purpose only - Direct Cytotoxicity Studies. Why not run this test on cells from the young and old groups?

Reply: Thank you so much for the query. BM-MSCs from middle-aged donors were used for cytotoxicity studies (direct and indirect). This decision was made due to the limited availability and low proliferation capacity of older donor BM-MSCs We have consented 14 older donors who have provided variable amounts of BM aspirates for expansion. Out of these, only 3 yielded sufficient cells needed for all the experiments. This is not uncommon given our previous experience with older donor BM-MSCs (Churchman et al., 2017). Therefore, the decision to use middle-aged donor MSCs for cytotoxicity experiments allowed us to preserve young and particularly, older donor BM-MSCs for the essential SNC osteogenesis experiments.

(3) Please add the sex of the donors. Also – is there any evidence in the literature for a different behaviour of BM-MSCs between males and females?

Reply: Thank you so much for the suggestion. The sex of the donors is now stated in relevant Materials and Methods section (lines 121-126).

Regarding the behaviour of BM-MSCs between males and females

Reply: Thank you for this valuable comment. Indeed, acknowledging possible sex differences is important but our present group sizes were insufficient to address this. As a response, we have added a following sentence to Discussion which reads:

Lines 525-528 Numerous studies have explored gender related differences in the proliferation and osteogenesis of BM-MSCs from humans (Seebach et al., 2007; Siegel et al., 2013). However, conflicting findings exist with some studies not observing such differences in humans (Ganguly et al., 2019; Jones and Schäfer, 2015).

Line 537-539 Considering gender differences, further investigation with male and female donors will establish the versatility of SNC in both genders.

(4) Experiments were carried out for 72 h, 7 days and 14 days. Why not use 3, 7 and 14 days? What is the importance of 72 hours?

Reply: Thank you so much for the comment. It has now been modified to 3 days

(5) Text on scale bar (figures 2 and 4) are too small to see. Please enlarge (see figures 5 and 6).

Reply: Thank you very much for the observation and suggestion. Scale bars have now been enlarged on figures 2 and 4 to make it clearer.

(6) Please always use the same significant decimal digits per parameters. For example: L288 “3.67 +/- 0.6” should be “3.7 +/- 0.6”; L290: “1.99 +/- 0.6” should be “2.0 +/- 0.6”.

Reply: Thank you very much for the comment. It is modified throughout the document

(7) Violin plots in figure 3: please explain in the legend what are the sashed and dotted lines (median, interquartile range?).

Reply: Thank you very much for the comment. The Figure 3 legend is modified as follows

Figure 3. Comparison of young and older donor BM-MSCs. (a) Representative images of methylene blue stained CFU-F plates. (b) Violin plots showing the colony area distribution and (c) colony integrated density of BM-MSC of young (n=3) and older donors (n=3).  Mann Whitney test **** - p < 0.0001. Data shown as violin plots with median (dashed lines) and interquartile range (dotted lines) values of colony area and colony integrated density. (d) PD time of young and older donor BM-MSCs. Unpaired t-test, Two-tailed, * - p < 0.05. Data shown as mean and standard deviation of the mean.

(8) can the authors add an image of the “SNC samples without the cells” (L318) to Figure 6 as this is the control. This will allow the reader to compare the results visually.

Reply: Thank you very much for the suggestion/advice.  This has now been incorporated in figure 6 as indicated below, and the corresponding results section has been modified (lines 358-362).

Figure 6. In vitro mineralization of BM-MSCs seeded on SNC: (A) Representative SEM-EDS images of SNC without cells, SNC with pooled young donor BM-MSCs and pooled older donor BM-MSCs. Red arrow indicated the presence of cells. Each element measured was superimposed onto the SEM image using pseudo colour, calcium (red), phosphorous (green) or silicon (blue). The scale bar represents 20µm; (B) Semi-quantitative analysis of the normalized mass (%) of calcium and phosphorus (n=3 images from different replicates). Data shown as mean and standard deviation of the mean.

(9) In several places in the manuscript the authors describe cell viability % above 100% (e.g., L261-270, L348). Can the authors explain shortly in the text what does it mean?

Reply: Thank you so much for the query. The cell viability was measured 24 hours after the addition of the extracts to the cells, the period of time not sufficient for MSCs to divide, based on our previous experience (Fragkakis et al., 2018). Hence the colour change reflected the metabolic activity of the attached live cells. On the other hand, cell proliferation was measured after 4 days of incubation of cells with the extracts, the period of time sufficient for MSCs to divide twice (Fragkakis et al., 2018). The values more than 100% reflect the enhanced metabolic activity in the cells compared to SM Medium (cell only) control. This could be due to mild pro-survival activity present in the extracts, but no specific trends for this enhancing effect were obvious. Overall, the data implied that extracts were non-cytotoxic and the cells remained metabolically active.

In response, the relevant section was re-written to read (lines 295-314)

Indirect cytotoxicity studies were carried out using sample extracts from SNC after incubation in SM media for 3 days (3D), 7 days (7D), and 14 days (14D). Cell viability of BM-MSCs was measured after 24 h of exposure, the period of time too short for cells to divide, and the cell proliferation was measured after 4 days of contact to the aforementioned extracts (the period sufficient for two MSC divisions) (Fragkakis et al., 2018). In the XTT assay, the yellow coloured XTT is reduced to orange coloured formazan by metabolically active cells and the orange colour developed is read at 450 nm using spectrophotometer; the metabolic activity therefore corresponds to the cell viability (24 h) or cell proliferation (4 days). Cell viability measurements (%) (Figure 2g) of BM-MSCs after contact with the extracts SNC 3D and SNC 14D were 115 ± 0.2% and 111 ± 3.1%, respectively, which were significantly (p < 0.001, p < 0.03) higher than SM Medium control (103 ± 3.2, indicating some pro-survival activity present in the extracts. Cell viability after contact with the sample SNC 7D was 103 ± 2.5%, which was not significantly different to SM media control. As expected, the 10% DMSO medium resulted in 42 ± 3% cell viability, significantly lower than SM Medium control (p<0.0001).

Cell proliferation measurements (%) (Figure 2h) of BM-MSCs after contact with SNC 3D, SNC 7D and SNC 14D were 88 ± 3.7%, 107 ± 8.8% and 93 ± 3.1% respectively, which were similar that of SM Medium control (100 ± 11.1%). As anticipated, the 10% DMSO medium displayed 44 ± 0.3% cell proliferation, which was significantly lower than SM Medium control (p<0.0001).

Specific comments:

- L42: “BM-MSCs”. This is the first time this abbreviation appears in the main text, please spell it out in full.

Reply: Thank you very much for the observation. It is modified in Line 45 as bone marrow mesenchymal stem/stromal cells (BM-MSCs)

- L45: “BM”. This is the first time this abbreviation appears in the main text, please spell it out in full.

Reply: Thank you very much for the comment. It is modified in Line 49.

- L46: “bone modeling”. Suggesting to change to “bone modeling and remodeling”.

Reply: Thank you very much for the comment. It is modified in Line 50.

- L66: “resorb-ability” should be “resorbability”

Reply: Thank you very much for the comment. It is modified in Line 70

- L103 “other ingredients” – please list them.

Reply: Thank you very much for the comment. It is included as following (line 107 -110)

SNC was formulated using shell nacre powder (72%) , SNLSM2 resin (12%) and other ingredients such as triethylene glycol dimethacrylate (12%) , fumed silica (3%), benzoyl peroxide- 0.7% (initiator), dimethylaminophenyl ethanol- 0.4% (activator), traces of butylated hydroxy toluene and 4-methoxy phenol (Wilson and Philipose Pampadykandathil, 2023). 

- L123: “SM”. This is the first time this abbreviation appears in the main text, please spell it out in full (Skim Milk?).

Reply: Thank you so much for the comment. It is included in the line 114-115.

- L281: what is the unit ID (5050 +/- 88 ID)? Colony integrated density? Please explain that in the text.

Reply: Thank you so much for the comment. It is modified in materials and methods Line 196 and in results Line321. 

- L289: ”p=0.0334” please change to P ≤ 0.05.

Reply: Thank you so much for the comment. It is modified in Line 306 and throughout the document.

Round 2

Reviewer 1 Report

Comments and Suggestions for Authors

The authors have satisfactorily addressed my concerns. One minor suggestion is to move Supplementary Figure 1 to the main manuscript if it is available.

Author Response

Dear Reviewer1,

Thank you so much for the comment. We have now included the supplementary figure 1 in the main manuscript as Figure 7B.

Reviewer 2 Report

Comments and Suggestions for Authors

The manuscript was considerably improved by the authors after this round of revisions. All the comments/suggestions were properly addressed.

Comments on the Quality of English Language

none

Author Response

Dear Reviewer 2,

Thank you so much for the appreciation.